# New Flavonoid Derivatives from *Melodorum fruticosum* and Their α-Glucosidase Inhibitory and Cytotoxic Activities

**DOI:** 10.3390/molecules27134023

**Published:** 2022-06-22

**Authors:** Lien T. M. Do, Jirapast Sichaem

**Affiliations:** 1Sai Gon University, Ho Chi Minh City 748355, Vietnam; 2Research Unit in Natural Products Chemistry and Bioactivities, Faculty of Science and Technology, Thammasat University Lampang Campus, Lampang 52190, Thailand; jirapast@tu.ac.th

**Keywords:** Annonaceae, *Melodorum fruticosum*, melodorones A–C, flavonoids, *α*-glucosidase inhibition, cytotoxicity

## Abstract

Three new flavonoid derivatives, melodorones A–C (**1**–**3**), together with four known compounds, tectochrysin (**4**), chrysin (**5**), onysilin (**6**), and pinocembrin (**7**), were isolated from the stem bark of *Melodorum fruticosum.* Their structures were determined on the basis of extensive spectroscopic methods, including NMR and HRESIMS, and by comparison with the literature. Compounds **1**–**7** were evaluated for their in vitro α-glucosidase inhibition and cytotoxicity against KB, Hep G2, and MCF7 cell lines. Among them, compound **1** exhibited the best activity against *α*-glucosidase and was superior to the positive control with an IC_50_ value of 2.59 μM. On the other hand, compound **1** showed moderate cytotoxicity toward KB, Hep G2, and MCF7 cell lines with the IC_50_ values of 23.5, 19.8, and 23.7 μM, respectively. These findings provided new evidence that the stem bark of *M. fruticosum* is a source of bioactive flavonoid derivatives that are highly valuable for medicinal development.

## 1. Introduction

*Melodorum**fruticosum* Lour (Annonaceae) is a shrub with fragrant yellow flowers distributed in South East Asia, more specifically indigenous to Laos, Cambodia, Thailand, and Vietnam. This plant has been used as a mild cardiac stimulant, a tonic, and as a hematinic to resolve dizziness [1]. Previous phytochemical studies on this plant have led to the isolation of terpenoids, aromatic compounds, butenolides, heptenoids, aporphine alkaloids, and flavonoids [2,3,4,5,6,7], some of which showed significant antioxidant, anti-inflammatory, cytotoxic, antiphytopathogenic, and antifungal activities [8,9,10,11]. However, there are no reports concerning α-glucosidase inhibition from *M. fruticosum*.

Flavonoids are a class of polyphenol secondary metabolites that are present in plants [12]. Some flavonoids, such as luteolin, kaempferol, baicalein, and apigenin, have hypoglycemic potential and have been reported as α-glucosidase inhibitors [13]. As an extension of a search for new flavonoids as α-glucosidase inhibitors from the *M. fruticosum*, three new flavonoid derivatives, melodorones A–C (**1**–**3**), together with four known compounds, tectochrysin (**4**) [14], chrysin (**5**) [15], onysilin (**6**) [16], and pinocembrin (**7**) [17] (Figure 1), were isolated from the dried stem bark of *M. fruticosum*. We describe here the isolation and structural elucidation of these compounds, together with the evaluations of their in vitro α-glucosidase inhibition and their cytotoxicity against different cancer cell lines (KB, Hep G2, and MCF7).

## 2. Results and Discussion

### 2.1. Structural Elucidation of the Isolates

Compound **1** was isolated as a colorless gum. The molecular formula C_21_H_20_O_5_ was determined by its HRESIMS, which displayed a molecular ion peak at *m*/*z* 351.1178 [M-H]^−^ (calcd. for C_21_H_19_O_5_ 351.1232), together with NMR data (Table 1, Appendix A). The characteristic resonances of a flavone core structure were evident from the ^1^H NMR data at δ_H_ 7.03 (s, H-3), and ^13^C NMR data at δ_C_ 163.4 (C-2), 104.9 (C-3), and 182.4 (C-4) [18]. The NMR data revealed the presence of one hydrogen-bonded hydroxy (δ_H_ 12.77), one methoxy (δ_H_ 3.94), and one prenyloxy (one oxymethylene protons at δ_H_ 4.46 (2H, d, *J* = 7.5 Hz), one olefinic proton at δ_H_ 5.45 (1H, m), two methyl protons at δ_H_ 1.71 (3H, s) and 1.63 (3H, s)) substituents. A singlet proton at δ_H_ 6.97 was assigned to aromatic proton H-8 of the ring A. The remaining proton resonances were typical of an unsubstituted ring B flavone at δ_H_ 8.11 (H-2**′**/6**′**) and 7.58–7.63 (H-3**′**/4**′**/5**′**). The HMBC correlation of H-1″ (δ_H_ 4.46) with C-6 (δ_C_ 130.6) allowed the placement of the prenyloxy moiety at C-6 (Figure 2). The methoxy group at δ_H_ 3.94 (3H, s) located at C-7 was proven by the HMBC correlation of the methoxy substituent protons with C-7 (δ_C_ 159.3). In turn, the HMBC correlations of 5-OH (δ_H_ 12.77) with C-5 (δ_C_ 152.8), C-6 (δ_C_ 130.6), and C-10 (δ_C_ 105.3) supported the position of the hydroxyl group at C-5. A careful comparison of the ^1^H and ^13^C NMR spectral data of **1** with baicalein [19] identified similar signals, distinguished by two hydroxyl groups at C-6 and C-7 (ring A) were replaced by the prenyloxy and methoxy substituents, respectively, in **1**. This deduction was strongly confirmed by the HSQC and HMBC correlations (Figure 2). From the aforementioned results, structure **1** was assigned as melodorone A.

Compound **2** was obtained as a colorless gum with [α]^25^_D_ −73.0 (c 0.01, CHCl_3_). Its molecular formula, C_21_H_22_O_5_, was established by its HREIMS, which showed a molecular ion peak at *m*/*z* 399.1448 [M+HCOO]^−^ (calcd. for C_22_H_23_O_7_ 399.1444). This was further supported by the ^1^H and ^13^C NMR spectral data, which displayed one oxymethine, one methoxy, one olefinic methine, two methylene, two methyl, six aromatic methine, and eight quaternary carbons (Table 1). The spectroscopic (^1^H and ^13^C NMR) patterns of **2** were very similar to those of **1** except for the presence of a single bond between C-2 and C-3 in the heterocyclic ring C. This flavanone moiety was proven by the ^1^H NMR spectrum, which showed an ABX spin system at δ_H_ 5.62 (1H, dd, *J* = 12.6, 3.0 Hz), 3.31 (1H, dd, *J* = 17.4, 13.2 Hz), and 2.82 (1H, dd, *J* = 16.8, 3.0 Hz), corresponding to H-2, H-3ax, and H-3eq, respectively [20]. The ^13^C NMR spectrum also confirmed the presence of a flavanone core structure because of the signals resonating at δ_C_ 42.2 (C-3), 78.7 (C-2), and 197.1 (C-4) of the flavanone ring C. The absolute configuration of C-2 was assigned as *S*-configuration through contrastive analysis of the optical rotation data of **2** ([α]^25^_D_ −73.0) with the known compound (−)-butin ([α]^26^_D_ −39.9) [21]. Based on the above spectral evidence, compound **2** was identified as a new flavanone and was named melodorone B.

Compound **3** was obtained as a colorless gum with [α]^25^_D_ −26.8 (c 0.01, CHCl_3_). The molecular formula C_20_H_20_O_4_ was determined by its HREIMS, which displayed a molecular ion peak at *m*/*z* 369.1363 [M+HCOO]^−^ (calcd. for C_21_H_21_O_6_ 369.1338). The ^13^C NMR spectrum (Table 1) showed three carbon signals at δ_C_ 198.7 (C-4), 78.4 (C-2), and 42.8 (C-3), which are frequently observed in the flavanone series [22]. The ^1^H NMR spectral data also justified the assignment of a flavanone moiety of **3** (Table 1). The presence of two hydroxyl substituents at C-5 (δ_C_ 148.3) and C-6 (δ_C_ 136.1) was identified by the HMBC correlations of 5-OH with C-5, C-6, and C-10 (δ_C_ 106.2), and of 6-OH with C-5, C-6, and C-7 (δ_C_ 139.5) (Appendix A). The ^1^H and ^13^C NMR data of **3** were nearly identical to those of (2*S*)-dihydrobaicalein [22], differing only in the resonance of the hydroxyl group was replaced by the resonance of a prenyl moiety at C-7 of ring A. This prenyl group was substituted at C-7, which was proven by the HMBC correlation of H-1″ (δ_H_ 3.26) with C-7. On the basis of these data, the structure of **3** was unambiguously established and named melodorone C.

### 2.2. α-Glucosidase Inhibitory Activity

Compounds **1**–**7** were assessed for their α-glucosidase inhibition, with acarbose as a positive control. The resulting IC_50_ values (Table 2) showed that all isolates, except **4**, **6**, and **7**, displayed stronger inhibitory effects on *α*-glucosidase than acarbose (IC_50_ 179 μM). Especially, compounds **1**–**3** and **5** exhibited *α*-glucosidase inhibition with IC_50_ values in the range of 2.59 to 4.00 μM, which were more strongly than acarbose. Among isolates, compound **1** revealed the most highly potent *α*-glucosidase inhibition with an IC_50_ value of 2.59 μM. Previous research reported that **5** inhibited α-glucosidase effectively with an IC_50_ value of 5.7 μM, while **4** and **7** showed no *α*-glucosidase inhibition [23], which was consistent with the findings of this study. The results indicated that the compounds from the stem bark of *M. fruticosum*, at least for **1**–**3** and **5**, were active *α*-glucosidase inhibitors that could be used for the treatment of diabetes mellitus.

### 2.3. Cytotoxic Activity against Human Cancer Cell Lines

All isolates were further assessed for their in vitro cytotoxicity against three human cancer cell lines (KB, Hep G2, and MCF7), using an MTT assay (Table 3). Compound **1** was found to exhibit moderate cytotoxicity toward all cancer cell lines, with IC_50_ values of 23.5, 19.8, and 23.7 μM, respectively. Compounds **2**, **4**, and **5** displayed weak cytotoxicity against all cancer cell lines, with IC_50_ values in the range of 32.0 to 80.9 μM, while compound **3** exhibited weak action against KB and Hep G2 cell lines, with IC_50_ values of 59.0 and 80.0 μM, respectively. Other compounds in this bioassay revealed no evident cytotoxicity (IC_50_ > 100). According to previous research, compound **4** inhibited moderate cytotoxicity against KB and MCF7 cell lines, with IC_50_ values of 55.27 and 16.7 µM, respectively [24], while **5** showed moderate cytotoxic effects on Hep G2 and MCF7 cell lines, with IC_50_ values of 50.55 and 37.20 µM, respectively [25,26]. On the other hand, compound **7** exhibited cytotoxicity against the MCF7 cell line with IC_50_ values of 226 and 108 µM for 48 and 72 h, respectively [27]. These results were similar and supported the findings of this study. Compounds **4** and **5** were previously reported to exhibit no cytotoxic effect on the normal human colon fibroblastic CCD-18co and normal human epidermal keratinocytes (NHEKs) cell lines [28,29]. Additionally, compounds **5** and **7** were also discovered to have no cytotoxicity against the normal Vero cell line [30]. However, the cytotoxicity of **1**–**3** and **6** against normal human cell lines is not available, but it is recommended to be determined in the future.

## 3. Materials and Methods

### 3.1. General Experimental Procedures

The NMR spectra were recorded using Bruker AvanceNEO 600 MHz and Bruker Avance III™ HD 500 MHz NMR spectrometers in DMSO-*d*_6_ (Merck, Darmstadt, Germany). The HRMS spectra were acquired on a X500_R_ QTOP model mass spectrometer (Sciex, Redwood City, CA, USA) and Dionex Ultimate 3000 HPLC system hyphenated with a QExactive Hybrid Quadrupole Orbitrap MS (Thermo Fisher Scientific, Waltham, MA, USA). Optical rotations were obtained with a A.KRÜSS Optronic P8000 polarimeter (KRÜSS, Hamburg, Germany). The IR data were measured on a Jasco 6600 FT-IR spectrometer using an ATR technique (Jasco, Tokyo, Japan). Silica gel (70–230 mesh, Merck, Darmstadt, Germany) and Sephadex LH-20 gel (GE Healthcare Bio-Sciences AB, Uppsala, Sweden) were used for column chromatography. TLC (silica gel 60 F254, Merck, Darmstadt, Germany) was used to monitor the fractions from column chromatography. *Saccharomyces cerevisiae* α-glucosidase, *p*-nitrophenyl-α-d-glucopyranoside (*p*NPG), acarbose, and dimethyl sulfoxide (DMSO) were purchased from Sigma-Aldrich (St. Louis, MO, USA).

### 3.2. Plant Material

*M. fruticosum* was collected in July 2017 from Lam Dong province, Vietnam. The material was authenticated by Dr. Dang Van Son. A voucher specimen (No US—A012) was deposited at the Herbarium of the Department of Organic Chemistry, Faculty of Chemistry, University of Science, National University–Ho Chi Minh City, Vietnam.

### 3.3. Extraction and Isolation

The dried and powdered stem bark of *M. fruticosum* (45.0 kg) was extracted with 95% EtOH (90 L × 3) at room temperature and concentrated under reduced pressure to give an EtOH crude extract (1.3 kg). This crude extract was suspended in water and partitioned with *n*-hexane and then EtOAc to yield *n*-hexane (45.0 g) and EtOAc (161.0 g) extracts. The *n*-hexane extract was fractionated by silica gel column chromatography (CC) eluted with *n*-hexane–EtOAc (90:10–0:10 *v*/*v*) and EtOAc–MeOH (10:0–0:10 *v*/*v*). The eluted fractions were combined into six fractions (HEX.1–HEX.6) on the basis of their TLC behavior. Fraction HEX.3 (6.5 g) was further separated by silica gel CC eluted with *n*-hexane–EtOAc (85:15 *v*/*v*) to give five subfractions (HEX.3.1–HEX.3.5). Subfraction HEX.3.2 (2.0 g) was purified by silica gel CC eluted with *n*-hexane–EtOAc (85:15 *v*/*v*) to give five subfractions (HEX.3.2.1–HEX.3.2.5). Subfraction HEX.3.2.2 was purified by silica gel CC eluted with *n*-hexane–EtOAc (85:15 *v*/*v*) to afford **3** (5.0 mg). Subfraction HEX.3.2.2 was applied to a Sephadex LH-20 gel CC (50.0 g) with CHCl_3_–MeOH (1:4 *v*/*v*) to obtain **6** (7.0 mg). Subfraction HEX.3.3 (0.8 g) was further purified by silica gel CC eluted with *n*-hexane–EtOAc (85:15 *v*/*v*), yielding four subfractions (HEX.3.3.1–HEX.3.3.4). Subfraction HEX.3.3.1 was further purified by silica gel CC eluted with *n*-hexane–EtOAc (85:15 *v*/*v*), followed by CC on Sephadex LH-20 gel eluted with CHCl_3_–MeOH (1:4 *v*/*v*) to afford **1** (6.8 mg) and **2** (5.4 mg). Subfraction HEX.3.3.4 was applied to a Sephadex LH-20 gel CC (50.0 g) with 100% MeOH to obtain **4** (10.7 mg). Subfraction HEX.3.4 (1.4 g) was separated by silica gel CC eluted with *n*-hexane–EtOAc (8:2 *v*/*v*), followed by CC on Sephadex LH-20 gel eluted with CHCl_3_–MeOH (1:4 *v*/*v*) to yield **5** (10.8 mg) and **7** (7.7 mg).

Melodorone A (**1**). Colorless gum. UV (CH_3_OH) _λmax_ (log ε) 275 (4.38), 295 (4.07), 356 (3.86) nm; IR (ATR) ν_max_ 3393, 2977, 1634, 1573, 1445, 1204, 1109 cm^−1^. HRESIMS m/z 351.1178 [M-H]^−^ (calcd. for C_21_H_19_O_5_ 351.1232); ^1^H NMR (DMSO-*d*_6_, 500 MHz) and ^13^C NMR (DMSO-*d*_6_, 125 MHz) see Table 1.

Melodorone B (**2**). Colorless gum. [α]^25^_D_ −73.0 (c 0.01, CHCl_3_); UV (CH_3_OH) _λmax_ (log ε) 275 (4.38), 295 (4.07), 356 (3.86) nm; IR (ATR) ν_max_ 3393, 2977, 1635, 1574, 1497, 1162, 1109 cm^−1^. HRESIMS *m*/*z* 399.1448 [M+HCOO]^−^ (calcd. for C_22_H_23_O_7_ 399.1444); ^1^H NMR (DMSO-*d*_6_, 600 MHz) and ^13^C NMR (DMSO-*d*_6_, 150 MHz) see Table 1.

Melodorone C (**3**). Colorless gum. [α]^25^_D_ −26.8 (c 0.01, CHCl_3_); UV (CH_3_OH) _λmax_ (log ε) 275 (4.38), 295 (4.07), 356 (3.86) nm; IR (ATR) ν_max_ 3392, 2977, 1634, 1450, 1217, 1122 cm^−1^. HRESIMS *m*/*z* 369.1363 [M+HCOO]^−^ (calcd. for C_21_H_21_O_6_ 369.1338); ^1^H NMR (DMSO-*d*_6_, 500 MHz) and ^13^C NMR (DMSO-*d*_6_, 125 MHz) see Table 1.

### 3.4. α-Glucosidase Inhibitory Assay

The α-glucosidase inhibition of all isolates was carried out according to a method adapted from a previous report [31]. Serial concentrations (2.0–256.0 μg/mL) of **1–7** and acarbose were prepared by dissolving in DMSO (400 mg/mL). Sodium phosphate buffer (100 mM, pH 6.8) was used to dissolve the α-glucosidase (0.4 U/mL) and substrate (2.5 mM *p*NPG). The substrate (40 μL) was added to the reaction mixture after the inhibitor (50 μL) was preincubated with α-glucosidase in 96-well plates at 37 °C for 10 min. A mixture without enzyme, sample compound, and acarbose served as blank, while the control contained only DMSO, enzyme, and substrate. The enzymatic reaction was carried out at 37 °C for 30 min and stopped by adding 0.2 M Na_2_CO_3_ (130 μL). Absorbance at 410 nm to measure enzyme activity was recorded on a BIOTEK reader. The assays were conducted in triplicates, with acarbose serving as a positive control. The IC_50_ values were calculated graphically using inhibition curves.

### 3.5. Cytotoxicity Assay

The cytotoxic evaluation of **1–7** and ellipticine against the growth of human epidermoid carcinoma (KB), human hepatocellular carcinoma (Hep G2), and human breast adenocarcinoma (MCF-7) cell lines was performed according to a previous procedure [32]. Ellipticine, a potent anticancer agent exhibiting multiple mechanisms of action, was used as the positive control [33,34,35]. The cancer cells were cultured in Dulbecco’s Modified Eagle’s Medium with 10% Fetal Bovine Serum, 1% penicillin and streptomycin, and 1% L-glutamine at 37 °C in a 5% CO_2_ environment. The tested compounds were added at concentrations ranging from 0.5 to 128 μg/mL by dissolving in DMSO (20 mg/mL) and incubated for a further 72 h in the identical condition. After the treatment, each well was filled with an MTT solution (10 μL, 5 mg/mL) of phosphate buffer, which was left to stand for 4 h until intracellular purple formazan crystals appeared. The MTT was removed and a 100 μL DMSO solution was added. The blank contained only a medium without any cells and MTT, as well as the solubilizing solution. The solution’s optical density (OD) at 540 nm was recorded on a BIOTEK reader. All experiments were carried out in triplicate for three independent experiments. IC_50_ values were computed graphically using inhibition curves.

## 4. Conclusions

The chemical investigation of the stem bark of *M. fruticosum* afforded the isolation of three unprecedented flavonoid derivatives (**1**–**3**), including one new flavone (**1**) and two new flavanones (**2** and **3**), along with four known compounds (**4**–**7**). All isolated were obtained from *M. fruticosum* for the first time. Compounds **1**–**3** and **5** exhibited highly potent inhibition against α-glucosidase and were superior to the positive agent. Furthermore, compounds **1**–**5** selectively showed in vitro cytotoxicity against KB, Hep G2, and MCF7 cell lines. The results of this study reveal that *M. fruticosum* stem bark is a highly useful source of bioactive flavonoid derivatives that should be explored further in the medical field.

## Figures and Tables

**Figure 1 molecules-27-04023-f001:**
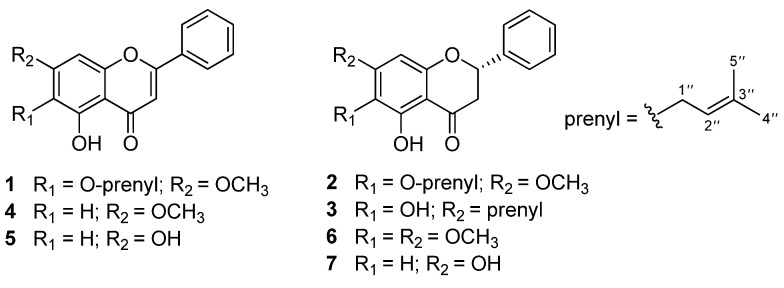
Chemical structures of **1**–**7**.

**Figure 2 molecules-27-04023-f002:**
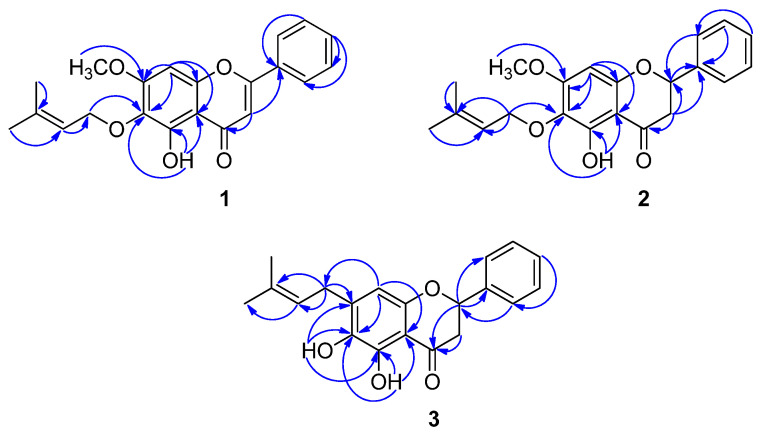
Key HMBC correlations of **1**–**3**.

**Table 1 molecules-27-04023-t001:** ^1^H and ^13^C NMR spectroscopic data of **1**–**3** recorded in DMSO-*d*_6_ (δ in ppm).

Position	1 ^a^		2 ^b^		3 ^a^	
	δ_H_ (*J* in Hz)	δ_C_	δ_H_ (*J* in Hz)	δ_C_	δ_H_ (*J* in Hz)	δ_C_
2		163.4	5.62, dd (12.6, 3.0)	78.7	5.55, dd (13.0, 3.0)	78.4
3ax	7.03, s	104.9	3.31, dd (17.4, 13.2)	42.2	3.31, overlapped	42.8
3eq			2.82, dd (16.8, 3.0)		2.83, dd (17.0, 3.0)	
4		182.4		197.1		198.7
5		152.8		154.4		148.3
6		130.6		128.3		136.1
7		159.3		161.1		139.5
8	6.97, s	91.6	6.30, s	92.0	6.27, s	106.4
9		152.3		158.3		153.0
10		105.3		102.5		106.2
1′		130.6		138.5		138.8
2′	8.11, dd (7.0, 1.5)	126.4	7.54, d (7.2)	126.6	7.53, dd (8.5, 1.5)	126.5
3′	7.58–7.63, m	129.1	7.44, dd (7.2, 7.2)	128.5	7.37–7.44, m	128.5
4′		132.1	7.40, m	128.6		128.4
5′		129.1	7.44, dd (7.2, 7.2)	128.5		128.5
6′	8.11, dd (7.0, 1.5)	126.4	7.54, d (7.2)	126.6	7.53, dd (8.5, 1.5)	126.5
1″	4.46, d (7.5)	68.3	4.35, d (7.2)	68.4	3.26, overlapped	28.7
2″	5.45, m	120.5	5.42, m	120.6	5.24, m	121.1
3″		137.4		137.1		132.7
4″	1.71, s	25.4	1.70, s	25.4	1.69, s	25.4
5″	1.63, s	17.7	1.61, s	17.7	1.65, s	17.6
5-OH	12.77, s		11.94, s		11.71, s	
6-OH					8.51, s	
7-OCH_3_	3.94, s	56.5	3.83, s	56.3		

^a^ NMR at 500/125 MHz. ^b^ NMR at 600/125 MHz.

**Table 2 molecules-27-04023-t002:** α-Glucosidase inhibition (IC_50 ±_ SD) of **1**–**7**.

Compound	IC_50_ (μM)
**1**	2.59 ± 0.15
**2**	3.33 ± 0.28
**3**	4.00 ± 0.20
**4**	>256
**5**	3.67 ± 0.25
**6**	192 ± 8.78
**7**	>256
Acarbose	179 ± 6.02

**Table 3 molecules-27-04023-t003:** Cytotoxicity of **1**–**7** against three human cancer cell lines.

Compound	IC_50_ ± SD (µM) ^a^		
	KB	Hep G2	MCF7
**1**	23.5 ± 1.1	19.8 ± 1.5	23.7 ± 2.0
**2**	62.1 ± 4.5	44.8 ± 4.0	73.7 ± 2.8
**3**	59.0 ± 2.5	80.0 ± 3.0	>100
**4**	64.7 ± 3.0	68.0 ± 2.5	80.9 ± 7.4
**5**	32.0 ± 0.0	71.4 ± 3.9	80.0 ± 4.5
**6**	>100	>100	>100
**7**	>100	>100	>100
Ellipticine ^b^	0.31 ± 0.05	0.33 ± 0.05	0.40 ± 0.05

^a^ Results are expressed as the means ± SD of three replicates. ^b^ Ellipticine was used as the positive control. Human epidermoid carcinoma (KB), human hepatocellular carcinoma (Hep G2), and human breast adenocarcinoma (MCF-7).

## Data Availability

All data supporting this study are available in the manuscript.

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
