# Peer review of "New Flavonoid Derivatives from Melodorum fruticosum and Their α-Glucosidase Inhibitory and Cytotoxic Activities"

_molecules, 2022, doi:10.3390/molecules27134023_

Round 1

Reviewer 1 Report

The document new flavonoid derivatives from Melodorum Fruticosum and their alfa-glucosidase inhibitory and cytotoxic activities is an interesting document of biological activities from new flavonoids of great interest, the document is well discussed and presented nevertheless some minimal comments need to be addressed prior publication.

At the end of the abstract section, the novelty of the study needs to be highlighted.

Additional information is required for the introduction section in order to improve the discussion, some information related to similar molecules and their characterization and application could be added.

The description of the general experimental procedures needs to be strengthened.

Most of the references are from 2015 to 2019 some new references could be added. 

Author Response

We would like to submit the revised manuscript “New flavonoid derivatives from Melodorum fruticosum and their α-glucosidase inhibitory and cytotoxic activities” to be considered for publication in Molecules. The amendments we have made according to the reviewer’s suggestion are enclosed in our revised manuscript. 

We hope that our changes are satisfactory, and we look forward to receiving your positive response.

Reviewer 2 Report

The manuscript by Do and Sichaem describes the isolation, characterization and evaluation of biological activity of three new flavonoid derivatives from the Asian medicinal plant M. fruticosum.

The manuscript is interesting, well-written and worth of publication; however, some issues have to be solved before:

1. Please, do not use capital letter fort the species name. The correct form is Melodorum fruticosum.

2. In the introduction section, please briefly state why you chose to test the isolated compounds on their anti-alpha-glucosidase potential? Also, explain why you used acarbose as a positive control in this assay.

3. Please explain why you used ellipticine as a positive control in cytotoxic assays, and cite some references regarding this compound. In the Materials&Methods section, doxorubicin is cited as a positive control for cytotoxicity assays – I assume this is a mistake?

4. Figure 1 is not cited in the text. Also, Table 2 is wrongly cited as Table 3. Please, correct.

5. Line 100: please delete “the” in the phrase: “….except 6 and 7, showed stronger the α-glucosidase….«

6. Line 115: “Compound 1” (not “Compounds 1”).

7. Materials&Methods: please, add the origin/producers of all the used chemicals, enzymes, cells and other material used in your research.

8. Chapter 3.4: “The inhibitor (50 µL) was preincubated with α-glucosidase…..«: please add the following information: (i) the names of used inhibitors (your compounds + positive control), (ii) the concentration (or the concentration range) of tested inhibitors, (iii) the solvent in which the inhibitors were dissolved prior to the addition to the test mixture, (iv) how you performed the blank experiments.

9. Chapter 3.5: please write (i) in which solvent were the compounds dissolved prior to the addition to the test, (ii) how you performed the blank experiments.

Author Response

(The authors gave the same response as above.)

Reviewer 3 Report

Authors isolated new natural products and also known compounds from Melodorum Fruticosum. The NMR identification is appreciated. The glucosidase and cytotoxic effect was tested. There are several questions and comments that should be addressed, and requires major revision.

1. Authors isolated three new compounds. How did they add the names? What is the reational behind that?

2. The glucosidase and cytotoxic effect of the known compounds isolated is well-known. Previous studies should be included to the introduction, and results should be compared with previous IC50s for example. 

3. Authors isolated couple of mg-s from 45 kg plant. Is this efficient? Is this significant? Based on the micromolar IC50s is it worth to work with so large amount of plant for so low amount of material?

4. In my opinion testing the single molecules is not giving the same result as it would by testing the mixture that is similar to that can be found in the plant. 

5. Authors are suggested to rationalize the choice of the control compounds. 

6. Authors are suggested to include the cytotoxicity against healthy cells as controls. 

Author Response

(The authors gave the same response as above.)

Round 2

Reviewer 3 Report

Authors answered and addressed the comments and questions and modified the manuscript accordingly. However, up to my opinion, these answers are not fully satisfying. 

The naming of the compounds is rational, I do agree with that. Although I understand that the comparison of previous biological studies are difficult to compare, I still recommend to include mentioning these data and to add that those might not be comparable. 

The efficacy of the isolation is still questioned. There are no isolated compounds available for further studies at this moment, but there was 45 kg plant applied a long time ago. Authors should add supporting information that confirms that this efficacy is usual in the case of isolating compounds from plants. 

I suggest the authors to look for literature data for healthy cell results, or at least mention that this data is not available, but it is recommended to add in the future.

Author Response

Dear Reviewer 3,

We would like to submit the revised manuscript “New flavonoid derivatives from Melodorum fruticosum and their α-glucosidase inhibitory and cytotoxic activities” to be considered for publication in Molecules. The amendments we have made according to the reviewer’s suggestion are enclosed in our revised manuscript.

We hope that our changes are satisfactory, and we look forward to receiving your positive response.
